# Sin3DM: Learning a Diffusion Model from a Single 3D Textured Shape

**Rundi Wu, Ruoshi Liu, Carl Vondrick, Changxi Zheng**
Columbia University
{rundi,rliu,vondrick,cxz}@cs.columbia.edu
https://sin3dm.github.io/

## Abstract

Synthesizing novel 3D models that resemble the input example has long been pursued by graphics artists and machine learning researchers. In this paper, we present Sin3DM, a diffusion model that learns the internal patch distribution from a single 3D textured shape and generates high-quality variations with fine geometry and texture details. Training a diffusion model directly in 3D would induce large memory and computational cost. Therefore, we first compress the input into a lower-dimensional latent space and then train a diffusion model on it. Specifically, we encode the input 3D textured shape into triplane feature maps that represent the signed distance and texture fields of the input. The denoising network of our diffusion model has a limited receptive field to avoid overfitting, and uses triplane-aware 2D convolution blocks to improve the result quality. Aside from randomly generating new samples, our model also facilitates applications such as retargeting, outpainting and local editing. Through extensive qualitative and quantitative evaluation, we show that our method outperforms prior methods in generation quality of 3D shapes.

## 1 Introduction

Creating novel 3D digital assets is challenging. It requires both technical skills and artistic sensibilities, and is often time-consuming and tedious. This motivates researchers to develop computer algorithms capable of generating new, diverse, and high-quality 3D models automatically. Over the past few years, deep generative models have demonstrated great promise for automatic 3D content creation (Achlioptas et al., 2018; Nash et al., 2020; Gao et al., 2022). More recently, diffusion models have proved particularly efficient for image generation and further pushed the frontier of 3D generation (Gupta et al., 2023; Wang et al., 2022b).

These generative models are typically trained on large datasets. However, collecting a large and diverse set of high-quality 3D data, with fine geometry and texture, is significantly more challenging than collecting 2D images. Today, publicly accessible 3D datasets (Chang et al., 2015; Deitke et al.,

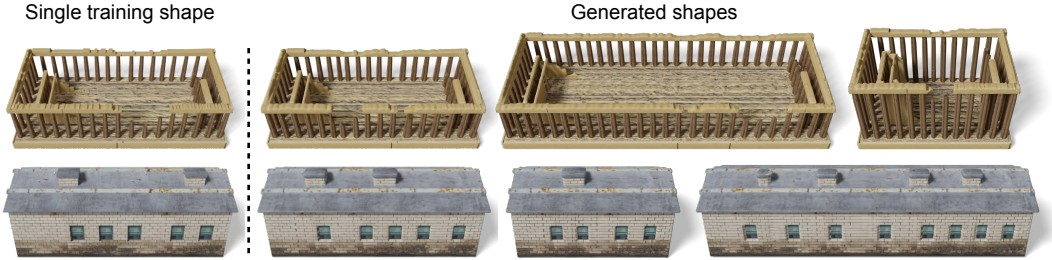

Single training shape          Generated shapes

Figure 1: Trained on a single 3D textured shape (left), Sin3DM is able to produce a diverse new samples, possibly of different sizes and aspect ratios. The generated shapes depict rich local variations with fine geometry and texture details, while retaining the global structure of the training example. Top: acropolis (choly kurd, 2021); bottom: industry house (Lukas carnota, 2015).

2022) remain orders of magnitude smaller than popular image datasets (Schuhmann et al., 2022), insufficient to train production-quality 3D generative models. In addition, many artistically designed 3D models possess unique structures and textures, which often have no more than one instance to learn from. In such cases, conventional data-driven techniques may fall short.

In this work, we present Sin3DM, a diffusion model that only trains on a single 3D textured shape. Once trained, our model is able to synthesize new, diverse and high-quality samples that locally resemble the training example. The outputs from our model can be converted to 3D meshes and UV-mapped textures (or even textures that describe physics-based rendering materials), which can be used directly in modern graphics engine such as Blender (Community, 2018) and Unreal Engine (Epic Games, 2019). We show example results in Fig. 1 and include more in Sec. 4. Our model also facilitates applications such as retargeting, outpainting and local editing.

We aim to train a diffusion model on a single 3D textured shape with locally similar patterns. Two key technical considerations must be taken into account. First, we need an expressive and memory-efficient 3D representation. Training a diffusion model simply on 3D grids would induce large memory and computational cost. Second, the receptive field of the diffusion model needs to be small, analogously to the use of patch discriminators in GAN-based approaches (Shaham et al., 2019). A small receptive field forces the model to capture local patch features.

The training process of our Sin3DM consists of two stages. We first train an autoencoder to compress the input 3D textured shape into triplane feature maps (Peng et al., 2020), which are three axis-aligned 2D feature maps. Together with the decoder, they implicitly represent the signed distance and texture fields of the input. Then we train a diffusion model on the triplane feature maps to learn the distribution of the latent features. Our denoising network is a 2D U-Net with only one-level of depth, whose receptive field is approximately $40\%$ of the feature map size. Furthermore, we enhance the generation quality by incorporating triplane-aware 2D convolution blocks, which consider the relation between triplane feature maps. At inference time, we generate new 3D textured shapes by sampling triplane feature maps using the diffusion model and subsequently decoding them with the triplane decoder.

To our best knowledge, Sin3DM is the first diffusion model trained on a single 3D textured shape. We demonstrate generation results on various 3D models of different types. We also compare to prior methods and baselines through quantitative evaluations, and show that our proposed approach achieves better quality.

## 2 RELATED WORK

**3D shape generation** Since the pioneering work by (Funkhouser et al., 2004), data-driven methods for 3D shape generation has attracted immense research interest. Early works in this direction follow a synthesis-by-analysis approach (Merrell, 2007; Bokeloh et al., 2010; Kalogerakis et al., 2012; Xu et al., 2012). After the introduction of deep generative networks such as GAN (Goodfellow et al., 2014), researchers start to develop deep generative models for 3D shapes. Most of the existing works primarily focus on generating 3D geometry of various representations, including voxels (Wu et al., 2016; Chen et al., 2021), point clouds (Achlioptas et al., 2018; Yang et al., 2019; Cai et al., 2020; Li et al., 2021), meshes (Nash et al., 2020; Pavllo et al., 2021), implicit fields (Chen & Zhang, 2019; Park et al., 2019; Mescheder et al., 2019; Liu et al., 2022; Liu & Vondrick, 2023), structural primitives (Li et al., 2017; Mo et al., 2019; Jones et al., 2020), and parametric models (Chen et al., 2020; Wu et al., 2021; Jayaraman et al., 2022). Some recent works take a step forward to generate 3D textured shapes (Gupta et al., 2023; Gao et al., 2022; Nichol et al., 2022; Jun & Nichol, 2023). All these methods rely on a large 3D dataset for training. Yet, collecting a high-quality 3D dataset is much more expensive than images, and many artistically designed shapes have unique structures that are hard to learn from a limited collection. Without the need of a large dataset, from merely a single 3D textured shape, our method is able to learn and generate its high-quality variations.

Another line of recent works (Poole et al., 2022; Lin et al., 2023; Wang et al., 2022a; Metzer et al., 2022; Liu et al., 2023) use gradient-based optimization to produce individual 3D models by leveraging differentiable rendering techniques (Mildenhall et al., 2020; Laine et al., 2020) and pretrained text-to-image generation models (Rombach et al., 2022). However, these methods have a long inference time due to the required per-sample optimization, and the results often show high saturation artifact. They are unable to generate fine variations of an input 3D example.

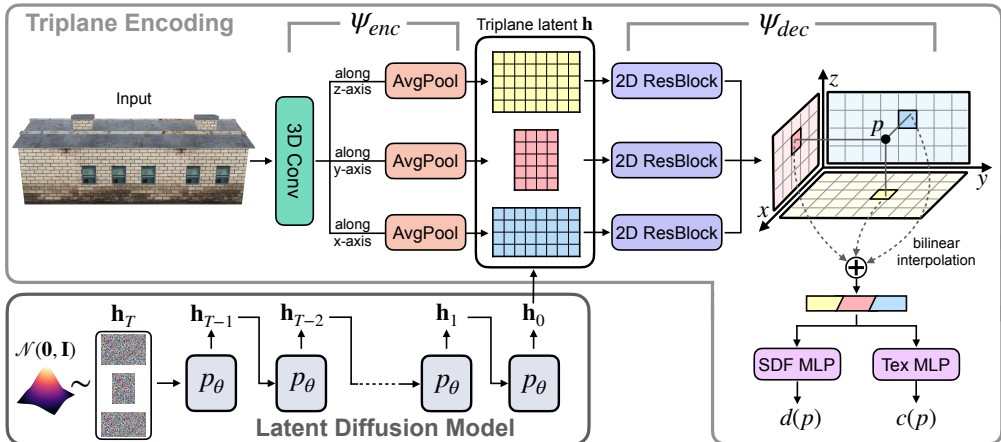

Figure 2: **Method overview.** Given an input 3D textured shape, we first train a triplane auto-encoder to compress it into an implicit triplane latent representation **h**. Then we train a latent diffusion model on it to learn the distribution of triplane features. See Fig. 3a for the structure of our denoising network $p_\theta$. At inference time, we sample a new triplane latent using the diffusion model and then decode it to a new 3D textured shape using the triplane decoder $\psi_{\text{dec}}$.

**Single instance generative models** The goal of single instance generative models is to learn the internal patch statistics from a single input instance and generate diverse new samples with similar local content. Texture synthesis is an important use case for such models and has been an active area of research for more than a decade Efros & Leung (1999); Han et al. (2008); Zhou et al. (2018); Niklasson et al. (2021); Rodriguez-Pardo & Garces (2022). Beyond stationary textures, the seminal work SinGAN (Shaham et al., 2019) explores this problem by training a hierarchy of patch GANs (Goodfellow et al., 2014) on an pyramid of a nature image. Many follow-up works improve upon it from various perspectives (Hinz et al., 2021; Shocher et al., 2019; Granot et al., 2021; Zhang et al., 2021). Some recent works use a diffusion model for single image generation by limiting the receptive field of the denoising network (Nikankin et al., 2022; Wang et al., 2022c) or constructing a multi-scale diffusion process (Kulikov et al., 2022). Our method is inspired by the above generative models trained on single image.

The idea of single image generation has been extended to other data domains, such as videos (Haim et al., 2021; Nikankin et al., 2022), audio (Greshler et al., 2021), character motions (Li et al., 2022; Raab et al., 2023), 3D shapes (Hertz et al., 2020; Wu & Zheng, 2022) and radiance fields (Karnewar et al., 2022; Son et al., 2022; Li et al., 2023). In particular, SSG (Wu & Zheng, 2022) is the most relevant prior work. It uses a multi-scale GAN architecture and trains a voxel pyramid of the input 3D shape. However, it only generates *un-textured* meshes (i.e., the geometry only) and the geometry quality is limited by the highest training resolution of the voxel grids. By encoding the input 3D textured shape into a neural implicit representation, our method is able to generate textured meshes with high resolutions for both geometry and texture.

Li et al. (2023) uses a patch matching approach to synthesize 3D scenes from a single example. It specifically focuses on 3D natural scenes represented as grid-based radiance fields (Plenoxels (Fridovich-Keil et al., 2022)), which cannot be easily re-lighted and we found its converted meshes are often broken.

**Diffusion models** Diffusion models (Sohl-Dickstein et al., 2015; Ho et al., 2020; Song et al., 2020) are a class of generative models that use a stochastic diffusion process to generate samples matching the real data distribution. Recent works with diffusion models achieved state-of-the-art performance for image generation (Rombach et al., 2022; Dhariwal & Nichol, 2021; Saharia et al., 2022; Ramesh et al., 2022). Following the remarkable success in image domain, researchers start to extend diffusion models to 3D (Zeng et al., 2022; Nichol et al., 2022; Cheng et al., 2022; Shue et al., 2022; Gupta et al., 2023; Anciukevičius et al., 2023; Karnewar et al., 2023) and obtain better performance than prior GAN-based methods. These works typically train diffusion models on a large scale 3D shape dataset such as ShapeNet (Chang et al., 2015) and Objaverse (Deitke et al., 2022; 2023). In contrast, we explore the diffusion model trained on a single 3D textured shape to capture the patch-level variations.

## 3 METHOD

**Overview** Sin3DM learns the internal patch distribution from a single textured 3D shape and generates high-quality variations. The core of our method is a denoising diffusion probabilistic model (DDPM) (Ho et al., 2020). The receptive field of the denoising network is designed to be small, analogously to the use of patch discriminators in GAN-based approaches (Shaham et al., 2019). With that, the trained diffusion model is able to produce patch-level variations while preserving the global structure (Wang et al., 2022c; Nikankin et al., 2022).

Directly training a diffusion model on a high resolution 3D volume is computationally demanding. To address this issue, we first compress the input textured 3D mesh into a compact latent space, and then apply diffusion model in this lower-dimensional space (see Fig. 2). Specifically, we encode the geometry and texture of the input mesh into an implicit triplane latent representation (Peng et al., 2020; Chan et al., 2021). Given the encoded triplane latent, we train a diffusion model on it with a denoising network composed of triplane-aware convolution blocks. After training, we can generate a new 3D textured mesh by decoding the triplane latent sampled from the diffusion model.

### 3.1 TRIPLANE LATENT REPRESENTATION

To train a diffusion model on a high resolution textured 3D mesh, we need a 3D representation that is expressive, compact and memory efficient. With such consideration, we adopt the triplane representation (Peng et al., 2020; Chan et al., 2021) to model the geometry and texture of the input 3D mesh. Specifically, we train an auto-encoder to compress the input into a triplane representation.

Given a textured 3D mesh $\mathcal{M}$, we first construct a 3D grid of size $H \times W \times D$ to serve as the input to the encoder. At each grid point $p$, we compute its signed distance $d(p)$ to the mesh surface and truncate the value by a threshold $\epsilon_d$. For points whose absolute signed distance falls within this threshold, we set their texture color $c(p) \in \mathbb{R}^3$ to be the same as the color of the nearest point on the mesh surface. For points outside the distance threshold, we assign their color values to be zero. After such process, we get a 3D grid $G_{\mathcal{M}} \in \mathbb{R}^{H \times W \times D \times 4}$ of truncated signed distance and texture values. In our experiments, we set $\max(H, W, D) = 256$.

Next, we use an encoder $\psi_{\text{enc}}$ to encode the 3D grid into a triplane latent representation $\mathbf{h} = \psi_{\text{enc}}(G_{\mathcal{M}})$, which consists of three axis-aligned 2D feature maps

$$\mathbf{h} = (\mathbf{h}_{xy}, \mathbf{h}_{xz}, \mathbf{h}_{yz}), \tag{1}$$

where $\mathbf{h}_{xy} \in \mathbb{R}^{C \times H' \times W'}$, $\mathbf{h}_{xz} \in \mathbb{R}^{C \times H' \times D'}$ and $\mathbf{h}_{yz} \in \mathbb{R}^{C \times W' \times D'}$, with $C$ being the number of channels. $H', W', D'$ are the spatial dimensions of the feature maps. The encoder $\psi_{\text{enc}}$ is composed of one 3D convolution layer and three average pooling layers for the three axes, as illustrated in Fig. 2.

The decoder $\psi_{\text{dec}}$ consists of three 2D ResNet blocks ($\psi_{\text{dec}}^{\text{xy}}$, $\psi_{\text{dec}}^{\text{xz}}$, $\psi_{\text{dec}}^{\text{yz}}$), and two separate MLP heads ($\psi_{\text{dec}}^{\text{geo}}$, $\psi_{\text{dec}}^{\text{tex}}$), for decoding the signed distances and texture colors, respectively. Consider a 3D position $p \in \mathbb{R}^3$. The decoder first refines the triplane latent $\mathbf{h}$ using 2D ResNet blocks and then gather the triplane features at three projected locations of $p$,

$$
\begin{aligned}
f_{xy} &= \text{interp}(\psi_{\text{dec}}^{\text{xy}}(\mathbf{h}_{xy}), \ p_{xy}), \\
f_{xz} &= \text{interp}(\psi_{\text{dec}}^{\text{xz}}(\mathbf{h}_{xz}), \ p_{xz}), \\
f_{yz} &= \text{interp}(\psi_{\text{dec}}^{\text{yz}}(\mathbf{h}_{yz}), \ p_{yz}),
\end{aligned}
\tag{2}
$$

where $\text{interp}(\cdot, q)$ performs bilinear interpolation of a 2D feature map at position $q$. The interpolated features are summed and fed into the MLP heads to predict the signed distance $\hat{d}$ and color $\hat{c}$,

$$
\begin{aligned}
\hat{d}(p) &= \psi_{\text{dec}}^{\text{geo}}(f_{xy} + f_{xz} + f_{yz}), \\
\hat{c}(p) &= \psi_{\text{dec}}^{\text{tex}}(f_{xy} + f_{xz} + f_{yz}).
\end{aligned}
\tag{3}
$$

We train the auto-encoder using a reconstruction loss

$$\mathcal{L}(\psi) = \mathbb{E}_{p \in \mathcal{P}}[|\hat{d}(p) - d(p)| + |\hat{c}(p) - c(p)|], \tag{4}$$

where $\mathcal{P}$ is a point set that contains all the grid points and 5 million points sampled near the mesh surface. $d(p)$ and $c(p)$ are the ground truth signed distance and color at position $p$, respectively.

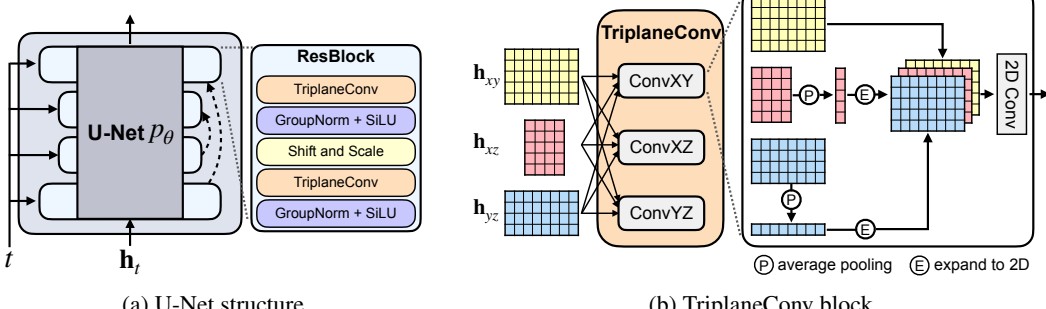

(a) U-Net structure                                  (b) TriplaneConv block

Figure 3: **Left: denoising network structure.** Our denoising network is a fully convolution U-Net composed of four ResBlocks and its bottleneck downsamples the input by 2. **Right: triplane-aware convolution block.** A TriplaneConv block considers the relation between triplane feature maps. Inside ConvXY, we apply axis-wise average pooling to $\mathbf{h}_{xz}$ and $\mathbf{h}_{yz}$, yielding two feature vectors, which are then expanded to the original 2D dimension by replicating along $y$(or $x$) axis. The two expanded 2D feature maps are concatenated with $\mathbf{h}_{xy}$ and fed into a regular 2D convolution layer.

Note that a seemingly simpler option is to follow an auto-decoder approach (Park et al., 2019), *i.e.*, optimize the triplane latent $\mathbf{h}$ directly without an encoder. However, we found the resulting triplane latent $\mathbf{h}$ to be noisy and less structured, making the subsequent diffusion model hard to train. An encoder naturally regularizes the latent space.

## 3.2 TRIPLANE LATENT DIFFUSION MODEL

After compressing the input into a compact triplane latent representation, we train a denoising diffusion probabilistic model (DDPM) (Ho et al., 2020) to learn the distribution of these latent features.

At a high level, a diffusion model is trained to reverse a Markovian forward process. Given a triplane latent $\mathbf{h}_0 = (\mathbf{h}_{xy}, \mathbf{h}_{xz}, \mathbf{h}_{yz})$, the forward process $q$ gradually adds Gaussian noise to the triplane features, according to a variance schedule $\{\beta_t\}_{t=0}^{T}$,

$$q(\mathbf{h}_t \,|\mathbf{h}_{t-1}) = \mathcal{N}(\mathbf{h}_t|\sqrt{1-\beta_t}\mathbf{h}_{t-1}, \beta_t\mathbf{I}). \tag{5}$$

The noised data at step $t$ can be directly sampled in a closed form solution $\mathbf{h}_t = \sqrt{\bar{\alpha}_t}\mathbf{h} + \sqrt{1-\bar{\alpha}_t}\boldsymbol{\epsilon}$, where $\boldsymbol{\epsilon}$ is random noise drawn from $\mathcal{N}(\mathbf{0}, \mathbf{I})$ and $\bar{\alpha}_t := \prod_{s=1}^{t} \alpha_s = \prod_{s=1}^{t}(1 - \beta_s)$. With a large enough $T$, $\mathbf{h}_T$ is approximately random noise drawn from $\mathcal{N}(\mathbf{0}, \mathbf{I})$.

A denoising network $p_\theta$ is trained to reverse the forward process. Due to the simplicity of the data distribution of a single example, instead of predicting the added noise $\boldsymbol{\epsilon}$, we choose to predict the clean input and thus train with the loss function

$$\mathcal{L}(\theta) = \mathbb{E}_{t\sim[1,T]}||\mathbf{h}_0 - p_\theta(\mathbf{h}_t, t)||_2^2. \tag{6}$$

**Denoising network structure.** A straight-forward option for the denoising network is to use the original U-Net structure from (Ho et al., 2020), treating the triplane latent as images. However, this leads to "overfitting", namely the model can only generate the same triplane latent as the input $\mathbf{h}_0$, without any variations. In addition, it does not consider the relations between the three axis-aligned feature maps. Therefore, we design our denoising network to have a limited receptive field to avoid "overfitting", and use triplane-aware 2D convolution to enhance coherent triplane generation.

Figure 3a illustrates the architecture of our denoising network. It's a fully-convolutional U-Net with only one-level of depth, whose receptive field covers roughly $40\%$ region of a triplane feature map of spatial size $128$. On the examples used in our experiments, we found that these parameter choices produce consistently plausible results and reasonable variations while keeping the input shape's global structure. In each ResBlock in our U-Net, we use a triplane-aware convolution block (see Fig. 3b), similar to the one introduced in (Wang et al., 2022b). It introduces cross-plane feature interaction by aggregating features via axis-aligned average pooling. As we will show in Sec. 4.3, it effectively improves the final generation quality.

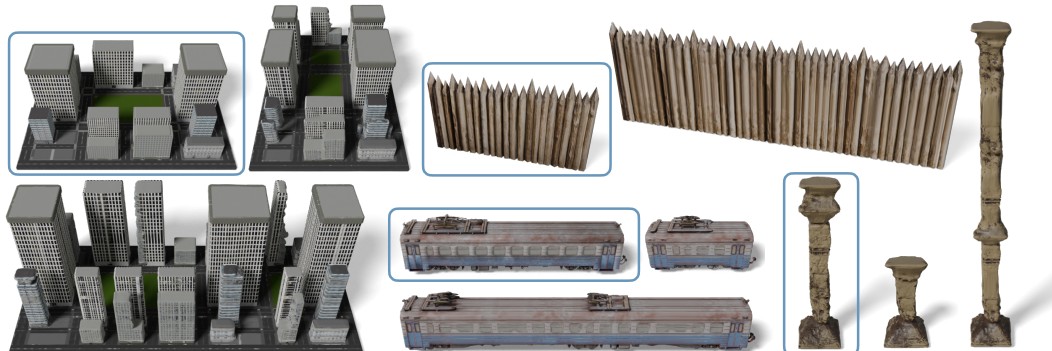

Figure 4: **Retargeting results.** By changing the spatial dimensions of the sampled Gaussian noise $\mathbf{h}_T$, we can resize the input to different sizes and aspect ratios. The training examples are labeled by blue boxes. From left to right, small town (Pedram Ashoori, 2020), wooden fence (REÂRCH Studio, 2018), train wagon (3ddominator, 2019) and antique pillar (oguzhnkr, 2017).

### 3.3 GENERATION

At inference time, we generate new 3D textured shape by first sampling new triplane latent using the diffusion model, and then decoding it using the triplane decoder $\psi_{\text{dec}}$. Starting from random Gaussian noise $\mathbf{h}_T \sim \mathcal{N}(\mathbf{0}, \mathbf{I})$, we follow the iterative denoising process (Sohl-Dickstein et al., 2015; Ho et al., 2020),

$$\mathbf{h}_{t-1} = \frac{\sqrt{\bar{\alpha}_{t-1}}\beta_t}{1 - \bar{\alpha}_t} p_\theta(\mathbf{h}_t, t) + \frac{\sqrt{\alpha_t}(1 - \bar{\alpha}_{t-1})}{1 - \bar{\alpha}_t}\mathbf{h}_t + \sigma_t\boldsymbol{\epsilon} \tag{7}$$

until $t = 1$. Here, $\boldsymbol{\epsilon} \sim \mathcal{N}(\mathbf{0}, \mathbf{I})$ for all but the last step ($\boldsymbol{\epsilon} = 0$ when $t = 1$) and $\sigma_t^2 = \beta_t$. After obtaining the sampled triplane latent $\mathbf{h}_0$, we first decode a signed distance grid at resolution 256, from which we extract the mesh using Marching Cubes (Lorensen & Cline, 1987). To get the 2D texture map, we use xatlas (Young, 2023) to warp the extracted 3D mesh onto a 2D plane and get the 2D texture coordinates for each mesh vertex. The 2D plane is then discritized into a $2024 \times 2048$ image. For each pixel that is covered by a warped mesh triangle, we obtain its corresponding 3D location via barycentric interpolation, and query the decoder $\psi_{\text{dec}}^{\text{tex}}$ to obtain its RGB color.

### 3.4 IMPLEMENTATION DETAILS

For all examples tested in the paper, we use the same set of hyperparameters. The input 3D grid has a resolution 256, *i.e.*, $\max(H, W, D) = 256$, and the signed distance threshold $\epsilon_d$ is set to $3/256$. The encoded triplane latent has a spatial resolution 128, *i.e.*, $\max(H', W', D') = 128$, and the number of channels $C = 12$.

We train the triplane auto-encoder for 25000 iterations using the AdamW optimizer (Loshchilov & Hutter, 2017) with an initial learning rate $5e{-}3$ and a batch size of $2^{16}$. The triplane latent diffusion model has a max time step $T = 1000$. We train it for 25000 iterations using the AdamW optimizer with an initial learning rate $5e{-}3$ and a batch size of 32. With the above settings, the training usually takes $2 \sim 3$ hours on an NVIDIA RTX A6000. Please see the Table 3 for detailed network configurations. We also include the source code in the supplementary materials.

## 4 EXPERIMENTS

### 4.1 QUALITATIVE RESULTS

We show a gallery of generated results in Fig. 8 and Fig. 9. We also show some shape retargeting results in Fig. 1 and Fig. 4, where we generate new shapes of different sizes and aspect ratios while keeping the local content. This is achieved by changing the spatial dimension of the sampled Gaussian noise $\mathbf{h}_T$. The generated shapes are able to preserve the global structure of the input example, while presenting reasonable local variations in terms of both geometry and texture. In the supplementary materials, we provide a webpage for interactive view of the generated 3D models.

Table 1: **Quantitative comparison.** ↓: lower is better; ↑: higher is better. We compare our method to SSG (Wu & Zheng, 2022) and Sin3DGenLi et al. (2023) in terms of geometry quality (G-Qual.), geometry diversity (G-Div.), texture quality (T-Qual.) and texture diversity (T-Div.). The last column is the average score over the 10 testing examples. Red: best score, Orange: second best score.

| Metrics | Methods | Acropolis | Canyon | Cliff Stone | Fight Pillar | House | Stairs | Small Town | Tree | Wall | Wood | Avg. |
|---------|---------|-----------|--------|-------------|--------------|-------|--------|------------|------|------|------|------|
| | | | | | | | | | | | | Examples |
| G-Qual. ↓ | Ours | 0.059 | 0.075 | 0.075 | 0.181 | 0.069 | 0.064 | 0.600 | 0.283 | 0.113 | 0.043 | 0.156 |
| | SSG | 0.055 | 0.077 | 0.150 | 0.247 | 0.017 | 0.179 | 0.928 | 0.376 | 0.244 | 0.111 | 0.238 |
| | Sin3DGen | 4.983 | 5.141 | 0.338 | 0.560 | 8.609 | 3.479 | 1.665 | 0.314 | 8.063 | 0.749 | 3.390 |
| G-Div. ↑ | Ours | 0.139 | 0.166 | 0.285 | 0.169 | 0.010 | 0.518 | 0.528 | 0.204 | 0.078 | 0.140 | 0.224 |
| | SSG | 0.091 | 0.218 | 0.342 | 0.175 | 0.008 | 0.514 | 0.463 | 0.040 | 0.101 | 0.134 | 0.209 |
| | Sin3DGen | 0.154 | 0.239 | 0.260 | 0.196 | 0.136 | 0.255 | 0.519 | 0.599 | 0.135 | 0.113 | 0.261 |
| T-Qual. ↓ | Ours | 0.024 | 0.032 | 0.006 | 0.012 | 0.019 | 0.114 | 0.051 | 0.007 | 0.013 | 0.004 | 0.028 |
| | SSG | 0.069 | 0.070 | 0.038 | 0.079 | 0.064 | 0.225 | 0.334 | 0.018 | 0.068 | 0.022 | 0.099 |
| | Sin3DGen | 0.039 | 0.042 | 0.009 | 0.017 | 0.026 | 0.020 | 0.135 | 0.027 | 0.045 | 0.004 | 0.036 |
| T-Div. ↑ | Ours | 0.066 | 0.218 | 0.132 | 0.151 | 0.053 | 0.116 | 0.255 | 0.234 | 0.048 | 0.056 | 0.133 |
| | SSG | 0.066 | 0.267 | 0.187 | 0.194 | 0.079 | 0.132 | 0.250 | 0.181 | 0.080 | 0.063 | 0.150 |
| | Sin3DGen | 0.061 | 0.232 | 0.117 | 0.119 | 0.041 | 0.151 | 0.289 | 0.137 | 0.059 | 0.028 | 0.123 |

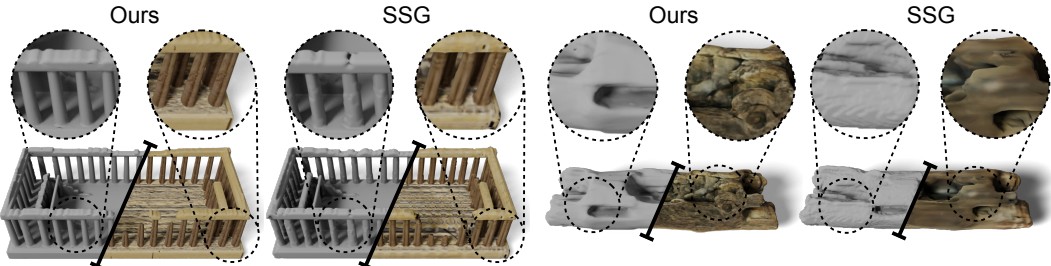

Figure 5: **Visual comparison.** We compare the generated results from our method and SSG (Wu & Zheng, 2022). The inputs of these two examples are shown in Fig. 8. Note that our mesh surfaces are much cleaner (see the zoomed-in columns), and our textures have much more details (see the zoomed-in wood surfaces). Please see Fig. 14 for the visual comparison to Sin3DGenLi et al. (2023).

## 4.2 COMPARISON

We compare our method against SSG (Wu & Zheng, 2022), a GAN-based single 3D shape generative model, and Sin3DGen Li et al. (2023), a recently proposed method that applies patch matching on the Plenoxels representation Fridovich-Keil et al. (2022). Since SSG only generates geometry, we extend it to support texture generation by adding 3 extra dimensions for RGB color at its generator's output layer. For a fair comparison, we train it on the same sampled 3D grid (resolution 256) that we used as our encoder input. In the Fig. 17 of the appendix, we show comparison to the original SSG on geometry generation only, by removing the texture component ($\psi_{\text{dec}}^{\text{tex}}$) in our model. For Sin3DGen, since it synthesize radiance fields that cannot be easily relighted, we generate its training data with the same lighting condition that we use for evaluation.

**Evaluation Metrics** To quantitatively evaluate the quality and diversity for both the geometry and texture of the generated 3D shapes, we adopt the following metrics. For geometry evaluation, we first voxelize the input shape and the generated shapes at resolution 128. *Geometry Quality (G-Qual.)* is measured against the input 3D shape using SSFID (single shape Fréchet Inception Distance) (Wu & Zheng, 2022). *Geometry Diversity (G-Div.)* is measured by calculating the pair-wise IoU distance $(1 - \text{IoU})$ among generated shapes.

For texture evaluation, we first render the input 3D model from 8 views at resolution 512. Each generated 3D model is then rendered from the same set of views. For the rendered images under each view, we compute the SIFID (Single Image Fréchet Inception Distance) (Shaham et al., 2019) against the image from the input model, and also compute the LPIPS metric (Zhang et al., 2018) between those images. *Texture Quality (T-Qual.)* is defined as the SIFID averaged over different views. Similarly, *Texture Diversity (T-Div.)* is defined as the averaged LPIPS metric.

We select 10 shapes in different categories as our testing examples. For each input example, we generate 50 outputs to calculate the above metrics. Formal definition of the above evaluation metrics are included in the Sec. C of the appendix.

Table 2: **Ablation study.** Metric values are averages over the 10 testing examples used in Table 1.

| | G-Qual. ↓ | G-Div. ↑ | T-Qual. ↓ | T-Div. ↑ |
|---|---|---|---|---|
| Ours | 0.156 | 0.224 | 0.028 | 0.133 |
| w/o triplaneConv | 0.347 | 0.310 | 0.068 | 0.171 |
| w/o encoder | 0.608 | 0.355 | 0.060 | 0.179 |
| $\epsilon$-prediction | 0.925 | 0.402 | 0.156 | 0.187 |

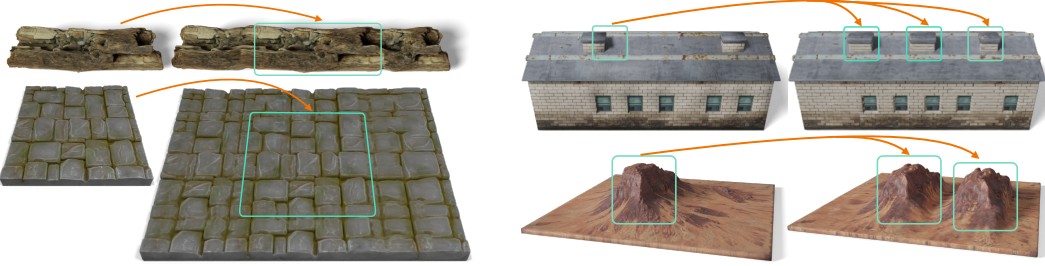

(a) Outpainting          (b) Patch duplication

Figure 6: **Controlled Generation.** Left: outpainting, which seamlessly extend the input 3D shape beyond its boundaries. Right: patch duplication, which copies a patch of the input to specified locations of the generated outputs. Wood (All-about-Blender-3D, 2020), stone tiles (skinny hands, 2021), house (Lukas carnota, 2015) and sandstone mountain (Marti David Rial, 2021).

**Results** We report the quantitative evaluation results in Table 1, and highlight the visual difference in Fig. 5 and Fig. 14. Compared to the baselines, our method obtains better scores for geometry and texture quality, while having similar scores for diversity. The generated shapes from SSG Wu & Zheng (2022) often have noisy geometry and blurry textures. This is largely because it is limited by the highest voxel grid resolution it trains on. Sin3DGen Li et al. (2023), based on Plenoxels radiance fields, often produces broken mesh surfaces and distorted textures (see Fig. 14). In contrast, our method is able to generate 3D shapes with high quality geometry and fine texture details.

### 4.3 ABLATION STUDY

We conduct ablation studies to validate several design choices of our method. Specifically, we compare our proposed method with the following variants:

*Ours (w/o triplaneConv)*, in which we do not use the triplane-aware convolution (Fig. 3b) in the denoising network. Instead, we simply use three separate 2D convolution layers for each plane, without considering the relation between triplane features.

*Ours (w/o encoder)*, in which we remove the triplane encoder $\psi_{\text{enc}}$ and fit the triplane latent in an auto-decoder fashion (Park et al., 2019). The resulting triplane latent is less structured, making the subsequent diffusion model hard to train.

*Ours ($\epsilon$-prediction)*, in which the diffusion model predicts the added noise $\epsilon$ instead of the clean input $\mathbf{h}_0$ (see Eqn. 6). In single example case, $\mathbf{h}_0$ is fixed and therefore easier to predict. Predicting the noise $\epsilon$ adds extra burden to the diffusion model.

As shown in Table 2, all these variants lead to lower scores for geometry and texture quality. The diversity scores increase at the cost of much lower result quality. Please refer to Fig. 12 for visual comparison. We also visually compare results of using different receptive field size in Fig. 13.

### 4.4 CONTROLLED GENERATION

Aside from randomly generating novel variations of the input 3D textured shape, we can also control the generation results by specifying a region of interest. Let $m$ denote a spatial binary mask indicating the region of interest. Our goal is to generate new samples such that the masked region $m$ stays as close as possible to our specified content $\mathbf{y}_0$ while the complementary region $(1 - m)$ are synthesized from random noise. Specifically, when applying the iterative denoising process (Eqn. 7), we replace the masked region with $\mathbf{y}_0$. That is, $\mathbf{h}_{t-1} \odot m \leftarrow \mathbf{y}_0 \odot m$, where $\odot$ is element-wise multiplication. To allow smooth transition around the boundaries, we adjust $m$ such that the borders

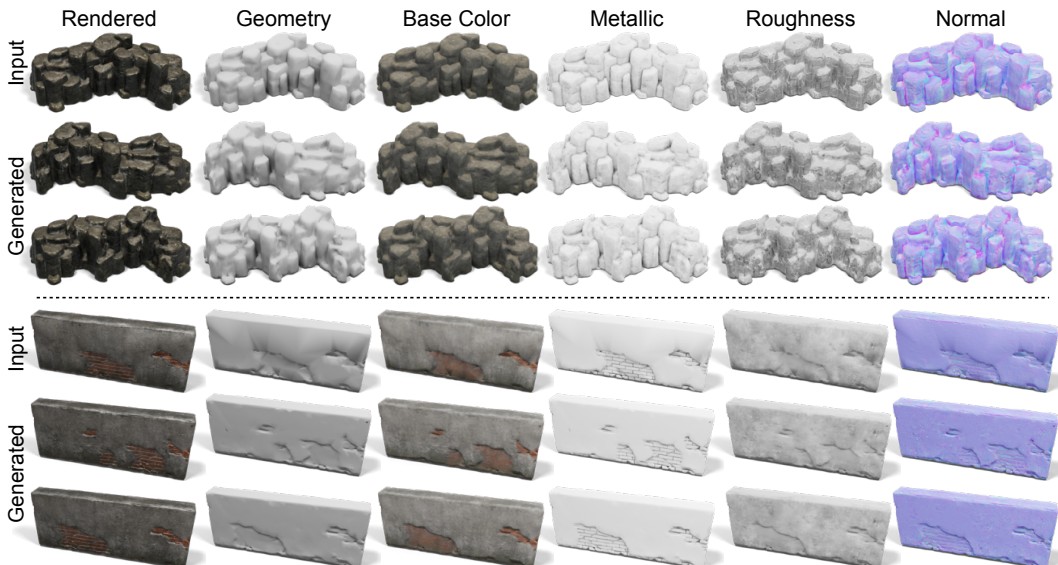

Figure 7: **Results on 3D models with PBR material.** Here we show two examples with PBR materials. For each example, we show the input model on top and two generated models below. Cliff stone (DJMaesen, 2021), damaged brick wall (Max Ramirez, 2016).

between $0$ and $1$ are linearly interpolated. Note that no re-training is needed and all changes are made at inference time.

We demonstrate two use cases in Fig. 6 and Fig. 15. 1) *Outpainting*, which seamlessly extends the input 3D textured shape beyond its boundaries. This is achieved by setting $\mathbf{y}_0$ to be the padded triplane latent $\mathbf{h}_0$ of the input (padded by zeros). The mask $m$ corresponds to the region occupied by the input. 2) *Patch duplication*, which copies the a patch of the input to the certain locations of the generated outputs where other parts are synthesized coherently. In the case, we take $\mathbf{h}_0$ and copy the corresponding features to get $\mathbf{y}_0$. The mask $m$ corresponds to the regions of the copied patches.

### 4.5 Supporting PBR Material

PBR (Physics-Based Rendering) materials are commonly used in modern graphics engines, as they provide more realistic rendering results. Our method can be easily extended to support input 3D shape with PBR material. In particular, we consider the material in terms of base color ($\mathbb{R}^3$), metallic ($\mathbb{R}$), roughness ($\mathbb{R}$) and normal ($\mathbb{R}^3$). Then, the input 3D grid has 9 channels in total (*i.e.*, $G_{\mathcal{M}} \in \mathbb{R}^{H \times W \times D \times 9}$). We add two additional MLP heads in the decoder — one for predicting metallic and roughness; the other for predicting normal. We demonstrate some examples in Fig. 7 and Fig. 10.

## 5 Discussion and Future Work

In this work, we present Sin3DM, a diffusion model that is trained on a single 3D textured shape with locally similar patterns. To reduce the memory and computational cost, we compress the input into triplane feature maps and then train a diffusion model to learn the distribution of latent features. With a small receptive field and triplane-aware convolutions, our trained model is able to synthesize faithful variations with intricate geometry and texture details.

While the use of triplane representation significantly reduces the memory and computational cost, we empirically observed that the generated variations primarily occur along three axis directions. Our method is also limited in controlling the trade-off between the generation quality and diverse, which is only possible by changing the receptive field size in our diffusion model. Generative models that learn from a single instance lack the ability to leverage prior knowledge from external datasets. Combing large pretrained models with single instance learning, possibly through fine-tuning (Ruiz et al., 2022; Zhang et al., 2022), is another interesting direction for synthesizing more diverse variations.

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

# A    RESULTS GALLERY

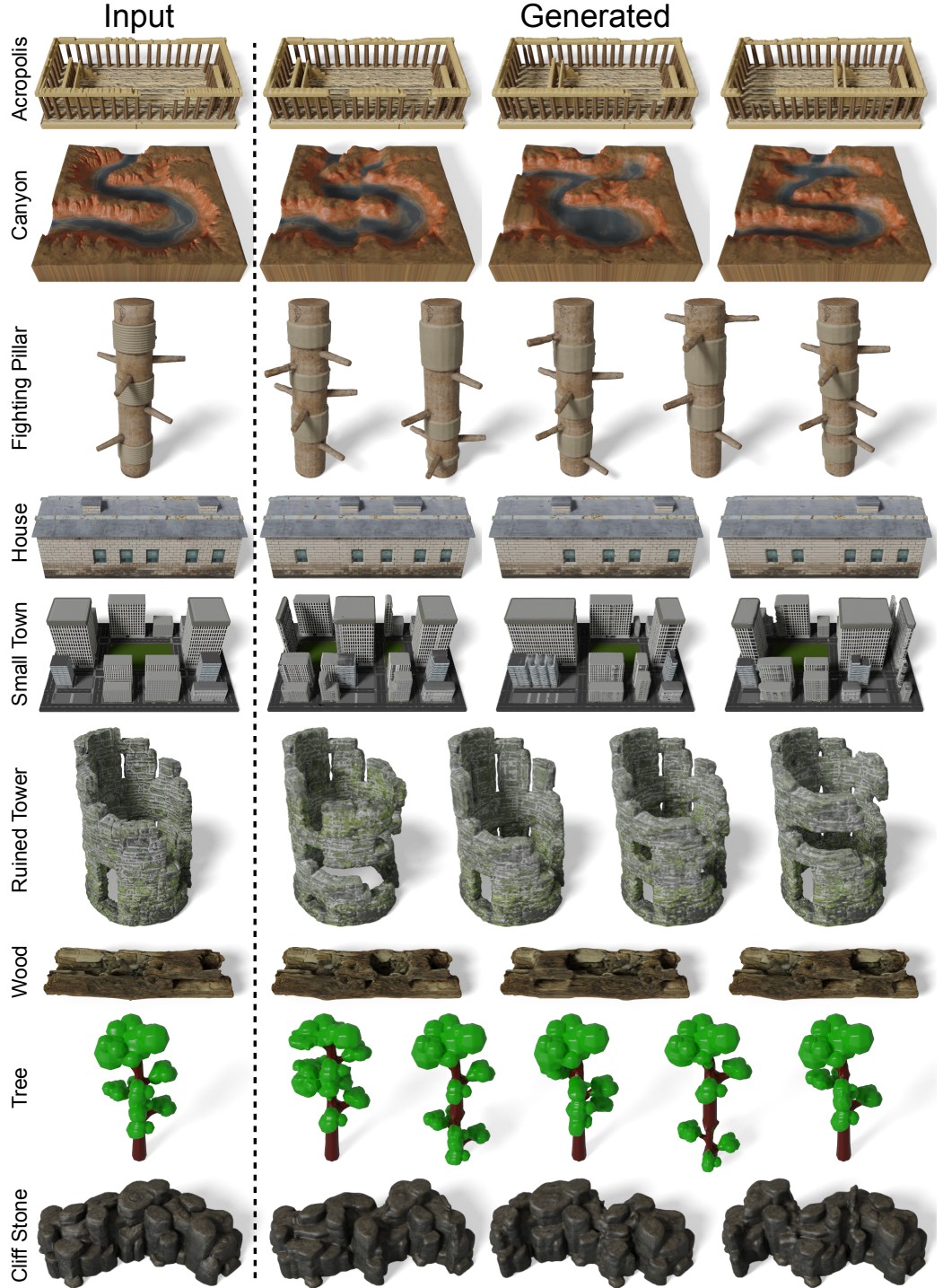

Figure 8: **Results gallery.** We show the input 3D textured shapes on the left, and several randomly generated samples on the right. From top to bottom, acropolis (choly kurd, 2021), canyon (Šimon Ustal, 2020), fighting pillar (ImpJive, 2021), house (Lukas carnota, 2015), small town (Pedram Ashoori, 2020), ruined tower (JB3D, 2019), wood (All-about-Blender-3D, 2020), tree (nikola, 2020) and cliff stone (DJMaesen, 2021).

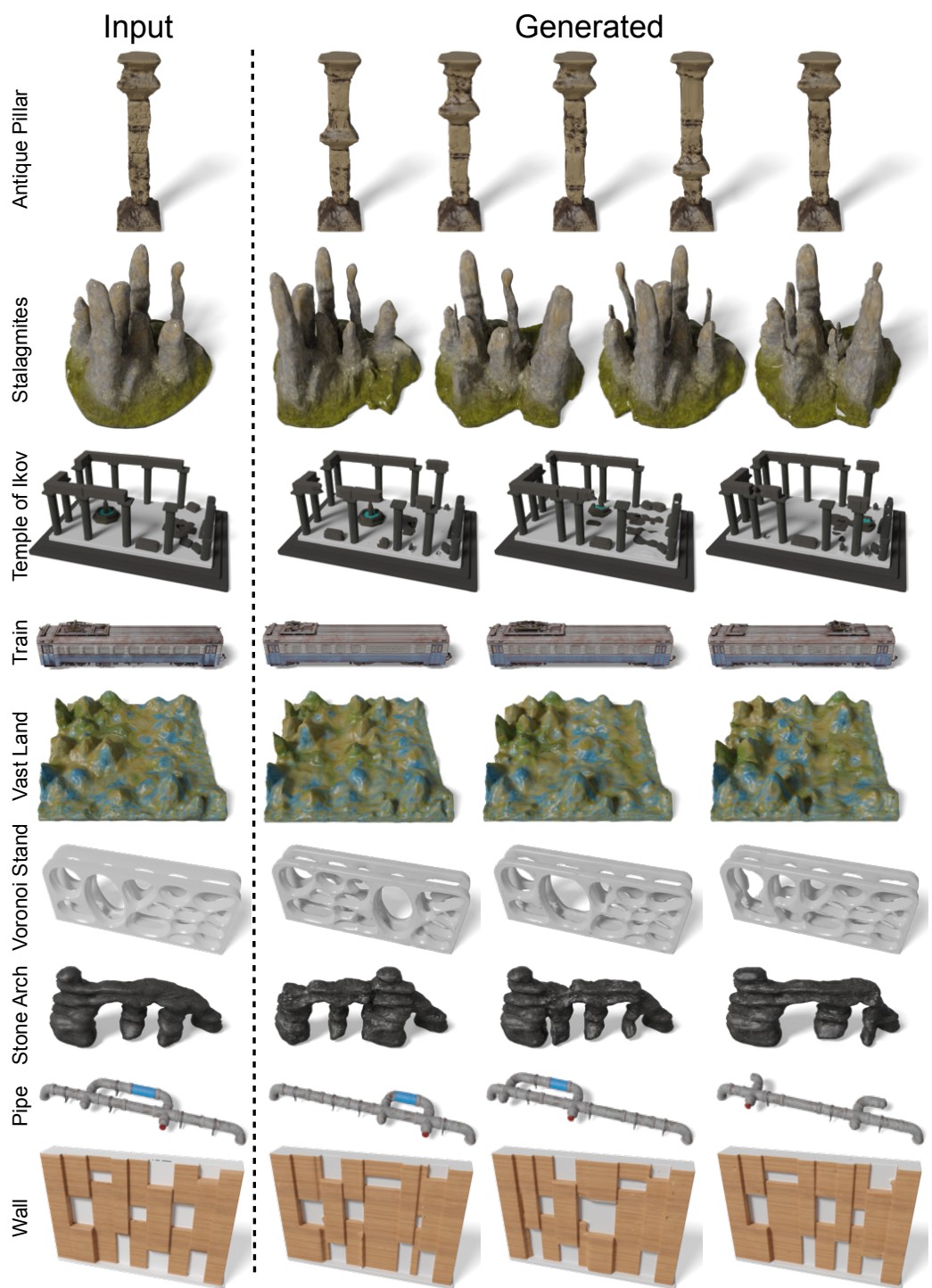

Figure 9: **Results gallery.** From top to bottom, antique pillar (oguzhnkr, 2017), stalagimites (Werniech van der Heever, 2019), temple of ikov (Laetitia Irata, 2019), train wagon (3ddominator, 2019), vast land (Shahriar Shahrabi, 2021), voronoi (Ahmad Riazi, 2013), stone arch (Werniech, 2022), pipe (SLASH / RENDAR, 2020), wall (stray, 2015).

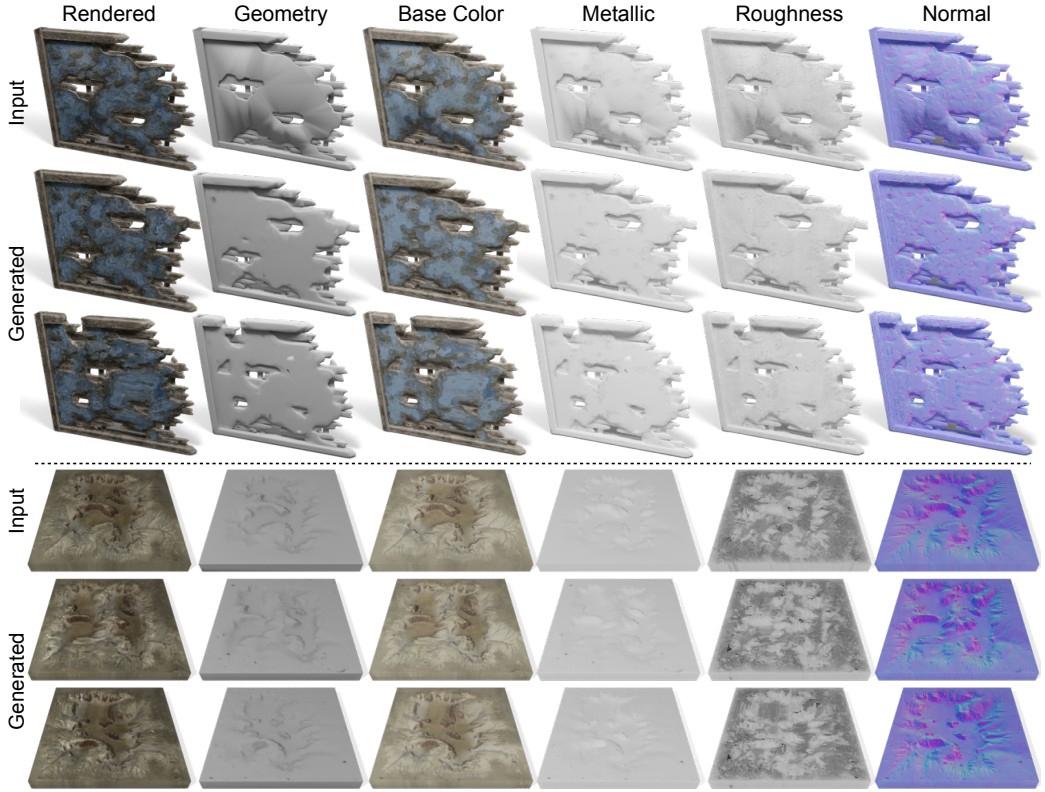

Figure 10: **More results on 3D models with PBR material.** Damaged wall (Max Ramirez, 2016), terrain (DJMaesen, 2018).

Table 3: **Network architectures.** `IN`: instance normalization layer. `GN`: group normalization layer. `Downsampling`: average pooling. `Upsampling`: bilinear interpolation with scaling factor 2. For the decoder MLP, we add a skip connection to the middle layer (Park et al., 2019).

| Module | Layers | Out channels | Kernel size | Stride |
|---|---|---|---|---|
| $\psi_{enc}$-conv | `Conv3D+IN+tanh` | 12 | 4 | 2 |
| $\psi_{dec}$-ResBlock | `Conv2D+IN+SiLU` | 64 | 5 | 1 |
| | `Conv2D` | 64 | 5 | 1 |
| | `Conv2D` (shortcut) | 64 | 1 | 1 |
| $\psi_{dec}$-MLP | 5 `[Linear+ReLU]` | 256 | - | - |
| | `Linear` | 1 or 3 | - | - |
| $p_\theta$ | `TriplaneConv` | 64 | 1 | 1 |
| | `TriplaneResBlock` | 64 | 3 | 1 |
| | `Downsampling` | - | 2 | 2 |
| | `TriplaneResBlock` | 128 | 3 | 1 |
| | `TriplaneResBlock` | 128 | 3 | 1 |
| | `Upsampling` | - | - | - |
| | `TriplaneResBlock` | 64 | 3 | 1 |
| | `GN+SiLU+TriplaneConv` | 12 | 1 | 1 |

# B  NETWORK ARCHITECTURES

# C  EVALUATION METRICS

For G-Qual. and G-Div., we refer readers to (Wu & Zheng, 2022) for the calculation of SSFID and diversity score based on IoU.

T-Qual. and T-Div. are computed as follows. We uniformly select 8 upper views (elevation angle $45°$) and render the textured meshes in Blender. Let $I_i(M)$ and $I_i(G_j)$ denote the rendered images at the $i-$th view, of the reference mesh $M$ and the generated mesh $G_j$, respectively. T-Qual. and T-Div. are then defined as

$$\text{T-Qual.} = \frac{1}{8}\sum_{i=1}^{8}[\frac{1}{n}\sum_{j=1}^{n}\text{SIFID}(I_i(M), I_i(G_j))],$$

$$\text{T-Div.} = \frac{1}{8}\sum_{i=1}^{8}[\frac{1}{k(k-1)}\sum_{j=1}^{n}\sum_{\substack{k=1\\k\neq j}}^{n}\text{LPIPS}(I_i(G_j), I_i(G_k))], \quad (8)$$

where we set $n = 50$. SIFID(Shaham et al., 2019) and LPIPS (Zhang et al., 2018) are distance measures.

## D MORE EVALUATIONS

Visual comparison for the ablation study (Sec. 4.3) is shown in Fig. 11 and Fig. 12. Ablation over different receptive field sizes is shown in Fig. 13. We also compare to SSG (Wu & Zheng, 2022) on geometry generation only, by removing the texture component ($\psi_{\text{dec}}^{\text{tex}}$) in our model. In Table 4 and Fig. 17, we show the quantitative and qualitative evaluation results on their 10 testing examples.

Table 4: **Quantitative comparison on geometry generation.** ↓: lower is better; ↑: higher is better. Please refer to (Wu & Zheng, 2022) for metrics definition. Numbers are average values over 10 testing examples.

|      | LP-IoU ↑ | LP-F-score ↑ | SSFID ↓ |
| --- | --- | --- | --- |
| Ours | 0.572 | 0.699 | 0.068 |
| SSG | 0.477 | 0.594 | 0.074 |

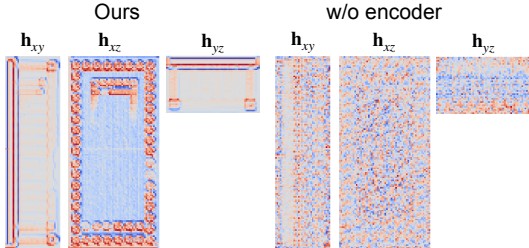

Figure 11: **Visualization of triplane feature maps.** Without an encoder, the learned triplane feature maps are noisy and less structured, which poses a difficult task for the subsequent diffusion model.

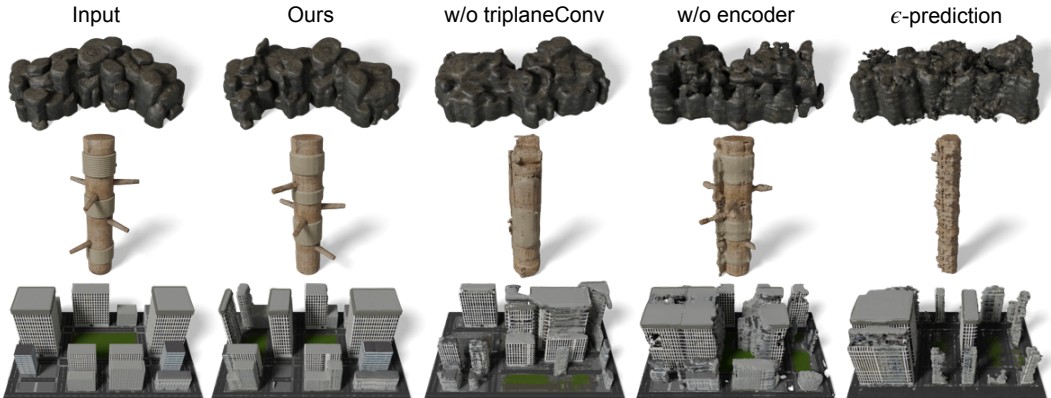

Figure 12: **Ablation study.** We show visual results from our method and compared variants.

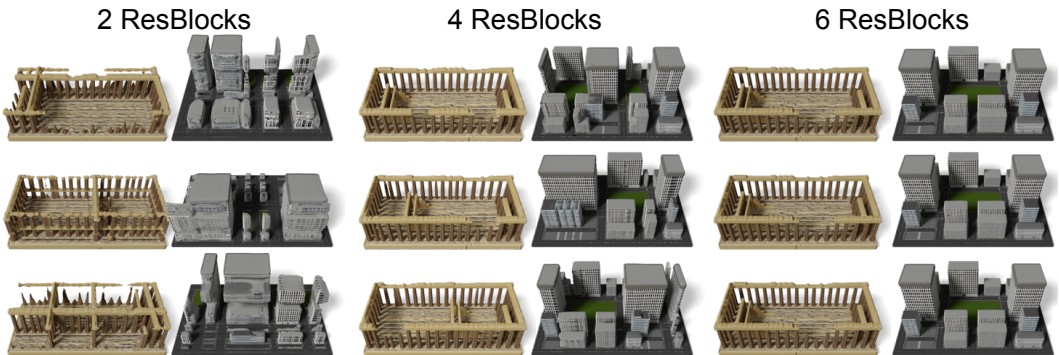

Figure 13: **Ablation over different receptive field sizes.**. We change the receptive field size by using different numbers of ResBlock in the U-Net. 2, 4, 6 ResBlocks have receptive fields that cover roughly 20%, 40%, 80% of the input. If the receptive field is too small, the generated shapes exhibit rich patch variations but fail to retain the global structure. If the receptive field is too large, the model overfits to the input example and is unable to produce variations.

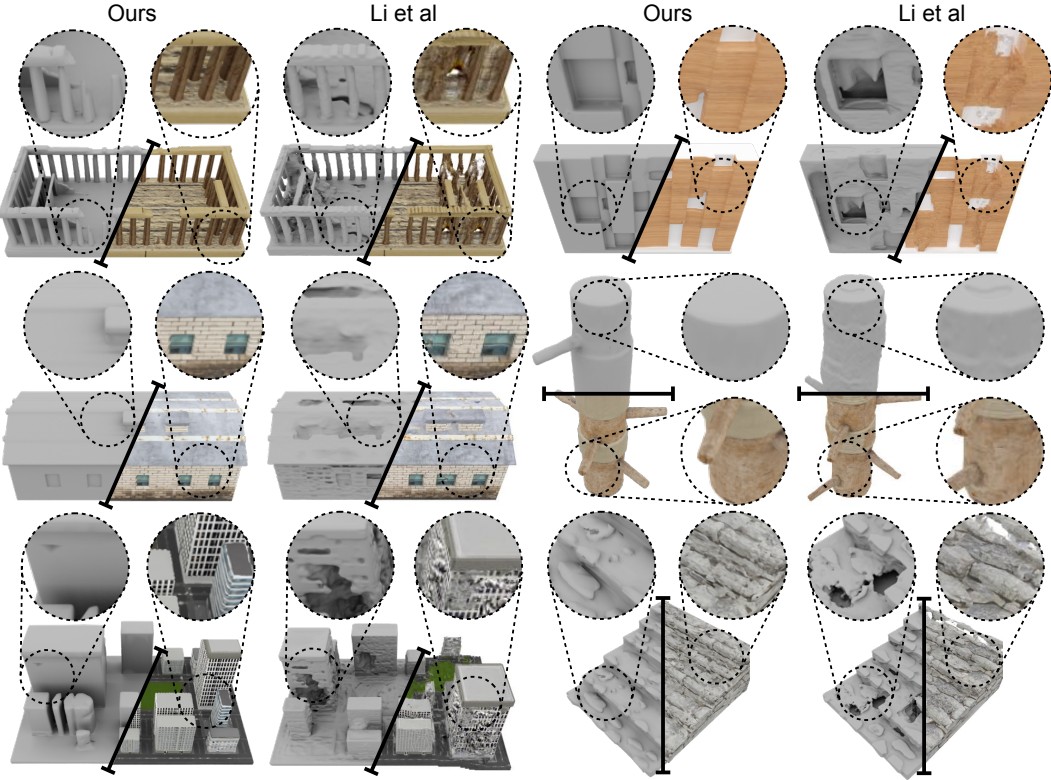

Figure 14: **Visual comparison to Sin3DGenLi et al. (2023).** The generated 3D shapes from Sin3DGen often depict bad geometry and sometimes their textures are distorted. In comparison, our generated shapes have high quality geometry with fine textures details.

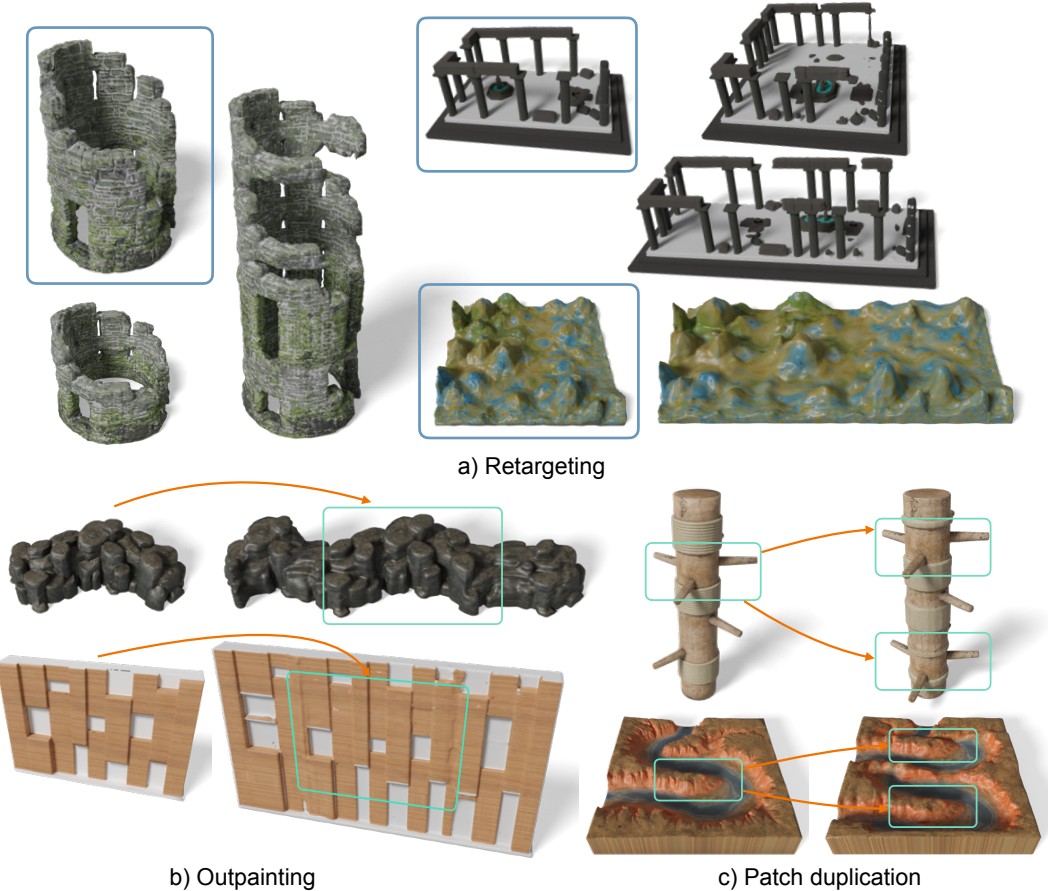

Figure 15: More examples for retargeting, outpainting and patch duplication.

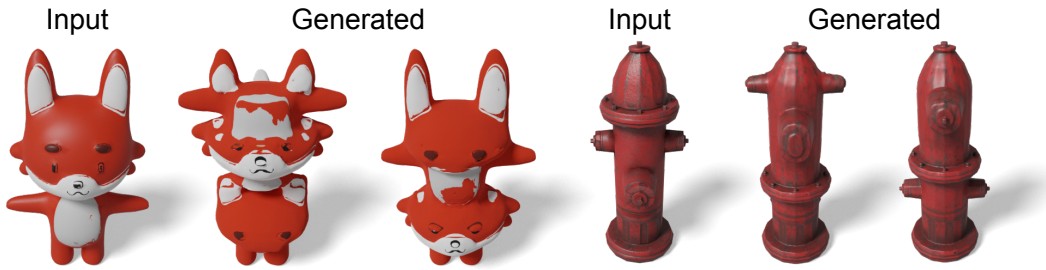

Figure 16: Failure cases. For inputs that have strong semantic structures, our method produces variations that do not necessarily maintain its semantics.

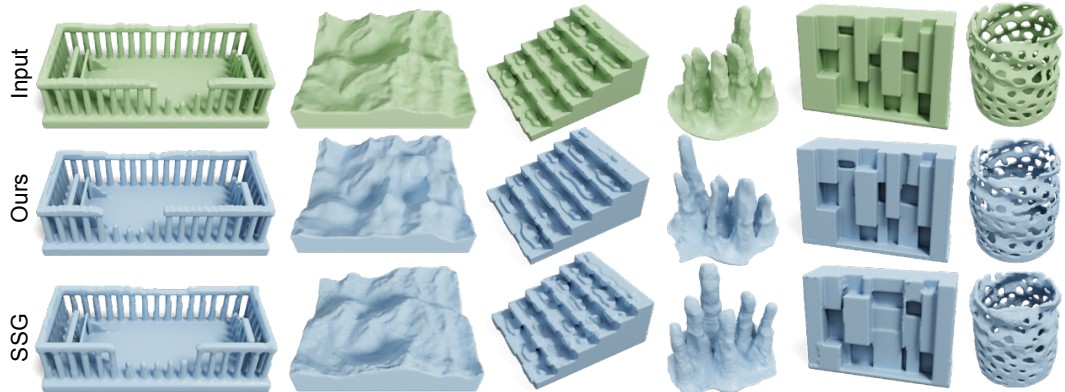

Figure 17: Visual comparison to SSG (Wu et al., 2016) on geometry generation. For most cases, our generated geometry are cleaner and sharper. But for examples with thin structures, our results are more likely to be broken (*e.g.*, the vase).

