# OpenReview forum: "Sin3DM: Learning a Diffusion Model from a Single 3D Textured Shape"
_ICLR.cc/2024/Conference — ICLR 2024 poster_

### Official Review · Reviewer_cGW2 · 2023-10-18

**Soundness:** 4 excellent
**Presentation:** 3 good
**Contribution:** 3 good
**Rating:** 8
**Confidence:** 4

**Summary:**

This paper introduces a novel generative model for generating 3D textured shapes, given a single example as input. Building upon recent work on diffusion models and tri-plane representations for shape representations, the proposed model can generate variations of the input shape. The method is evaluated across a variety of metrics of quality and diversity and compared to previous work on this topic. An ablation study is included, showing the impact of each of the components of the model. Finally, important applications are shown, including generating 3D shapes with BRDF properties, as well as different edition possibilities.

**Strengths:**

- This paper introduces a novel generative model for 3D textured shape generation, capable of synthesizing unseen variations of a 3D texture shape given a single example as input. Generative models trained on a single example (image, texture, video, etc) are a long-standing problem in the literature, and this paper proposes, to the best of my knowledge, the first method for 3D textured shapes.
- The proposed method is sound, the design decisions are solid and well-justified, and borrow from the literature of both neural rendering and single image diffusion models in interesting ways.
- Some ideas in this paper may be valuable for many downstream applications, including generative models trained on datasets of 3D textured shapes, or for dreamfusion-like models, where the representation that is learned is a radiance field.
- The quantitative evaluation proposed in this paper is comprehensive, measuring not only quality but also diversity.
- The applications proposed in this paper demonstrate that the method can be used for controllable generation and edition. Further, it can also synthesize BRDF properties, which makes this model applicable for downstream computer graphics pipelines.
- The paper is well structured and it is mostly well written. The quality of the figures is very high.
- The triplane convolution is smart and I believe it amy have an impact on many future works.
- The supplementary material provides valuable insights on the quality of the results and the impact of different individual components.
- Extensive implementation details are provided, and code is included in the submission, increasing the paper's reproducibility.

**Weaknesses:**

- The paper writing is sometimes overly convoluted and the same arguments are repeated too many times. In my opinion, this makes the paper at times a bit hard to read. (See the Questions section).
- In some cases, showing more examples could have been benefitial to this paper.
- While limitations are somewhat adequately discussed, it would have been interesting to see failure cases.
- The applications (4.4 and 4.5) sections are limited in the amount of examples shown and it makes it hard to fully understand what level of control is actually provided to the user.
- The related work section can be, in my opinion, improved. In particular, I believe that this paper should mention an important use-case for single-sample generative models, which is texture synthesis. This is even more important as this paper is working with Textured 3D shapes, and most of the examples shown contain repetitions (eg bricks or windows). I suggest including important work like "Non-stationary texture synthesis by adversarial expansion", Zho et al., Siggraph 2018; "“Self-organising textures", Niklasson et al. 2021; and "SeamlessGAN: Self-Supervised Synthesis of Tileable Texture Maps", Rodriguez-Pardo et al. TVCJ 2022.
- Some components of the model are detrimental to the diversity of the generated samples. This is a traditional trade-off in many generative models, where increased diversity can lead to less realistic outputs. However, I believe that this paper leans heavily towards realism, hindering diversity. I believe that some level of control on this tradeoff should be allowed to the user, and changing the receptive field of the model is, in my eyes, the only possible way to achieve this. This could be discussed in the limitations section or some future work could be hinted to address this.

**Questions:**

- What are the results of this method with unstructured shapes? Most of the results shown contain repeating patterns (like columns, windows or bricks). What would happen with a human 3D shape, for instance?
- Can this method generate tileable texture shapes? For example, it would be valuable to generate a tileable version of the shape in Figure 6 (a).
- Regarding the resolution of the method, it is mentioned that the maximum resolution across one dimension is 256 for the input shape, and that the receptive field is 128, while it is also mentioned that the receptive field is 40% of the input resolution. Could the authors clarify this?
- Does this model allow for texture transfer or for structural analogies? See "Non-Stationary Texture Synthesis by Adversarial Expansion", "Drop the GAN" or "Neural Photometry-guided Visual Attribute Transfer" for examples of this. It would be very powerful to be able to condition the generation of one or more of the tri-planes with a structure from another image.
- Is there a middle ground between using and not using the $\epsilon$-prediction? I am concerned that this introduces a trade-off between diversity and quality that may be limiting the impact of the proposed method.
- Can the authors show more examples of results of this method? I think it would be valuable to see more results on controlled generation, retargeting and PBR generation.
- What is the impact of using only a 2-level U-Net for the denoising, rather than

Writing improvement suggestions:
- I suggest the author mention the need for relightable assets less frequently. I think the introduction motivates the problem well, however, it is mentioned more times during the paper, hindering its readability.
- On the first paragraph of the introduction: "often time-consuming and tedious", as the current text is written, is referring to "artistic sensibilities", which I would disagree that are "time-consuming and tedious". I understand where the authors are going with this argument, however, the current phrasing is a bit confusing.

**Details Of Ethics Concerns:**

The authors have mentioned that "I certify that there is no URL (e.g., github page) that could be used to find authors' identity." on their submission. However, I came across this URL which is not anonymized: https://github.com/Sin3DM/Sin3DM . I am not sure this complies with ICLR code of conduct and I would ask the ACs to look into this. My review has not been influenced by this in any way.

---

> ### Author Response · Authors · 2023-11-19
> **Response from authors**
>
> __Q: Paper writing.__
> __A:__ Thanks for the suggestion. We have fixed the typos and mentioned less frequently the need for relightable assets in the revised paper. These changes are highlighted with color in the revised paper.
>
> __Q: Failure cases. What are the results of this method for shapes without repeated patterns?__
> __A:__ With a single input shape, our method learns its local patterns. Thus, for an input shape that has a highly semantic structure but no repeated patterns, e.g., a 3D character, the output may lose the semantic and thus looks unrealistic. In the revised paper, we show failure cases in Figure 16 of the appendix.
>
> __Q: Need more examples on controlled generation and retargeting.__
> __A:__ In the revised paper, we show more examples of retargeting and controlled generation in Figure 15 of the appendix.
>
> __Q: Improve the related work section by discussing prior papers on texture synthesis.__
> __A:__ Thanks for the suggestion and we have improved the related work section accordingly in the revised paper.
>
> __Q: The trade-off between diversity and quality can only be controlled by changing the receptive field size. This could be discussed in the limitations section.__
> __A:__ We acknowledge this limitation and include a discussion on it in the revised paper.
>
> __Q: Can this method generate tileable texture shapes?__
> __A:__ No, we didn’t come up with a way to enforce the output to be tileable.
>
> __Q: Clarify the receptive field size.__
> __A:__ Sorry for this confusion. The size of the receptive field is 54, which is roughly 40% of the triplane size 128.
>
> __Q: Does this model allow for texture transfer or for structural analogies?__
> __A:__ No. The geometry and texture is coupled in our triplane latent space, which makes it difficult to do texture transfer or structural analogies. This is left for future work.
>
> __Q: Is there a middle ground between using and not using the epsilon-prediction?__
> __A:__ In the diffusion model literature, there is another alternative objective ‘v-precition’ which is formulated as a combination of $x_0$ and $\epsilon$, $v=\alpha_t \epsilon - \sigma_t x_0$. This might be a middle ground that gives better a trade-off.
>
> __Q: Ethics concerns.__
> __A:__ We appreciate the reviewer for bringing up the ethics concern. However, we did NOT include that url anywhere in the submission package. It is our understanding according to ICLR submission policy  (https://iclr.cc/Conferences/2024/CallForPapers) that an arxiv preprint is allowed. We therefore believe we did NOT violate the ICLR code of conduct. In this year and past years’ ICLR review periods, we have seen many submissions (and accepted papers) that follow a similar way of disseminating their works, namely, having non-anonymous public web-pages while submitting fully anonymously.

---

### Official Review · Reviewer_s6cL · 2023-10-30

**Soundness:** 3 good
**Presentation:** 3 good
**Contribution:** 3 good
**Rating:** 5
**Confidence:** 3

**Summary:**

This paper aims at training a diffusion model to generate tri-plane that could be decoded into 3D objects. It first compress the 3D meshes into tri-planes by training a auto-encoder. Next, it trains a diffusion model to generate the encoded triplane. By altering the input noise, the model could achieve retargeting, outpainting and local editing.

**Strengths:**

1. The edited geometry after altering the noise is satisfactory. It has rich local variation while retaining the global structure.
2. The writing is clear.

**Weaknesses:**

1. After training, the model could only generate 3D contents with the same global structure. Since we already have the basic 3D mesh at the beginning, it seems that the application is a bit narrow and limited. Besides, achieving this goal needs to train two models, an auto-encoder and a diffusion model. The outcome does not seem to match the effort.
2. The novelty of this work lies at the usage of tri-plane as the input of diffusion model and the introduction of tri-plane convolution. Both aspects do not offer a solid contribution. Using tri-plane as a compact 3D representation has been explore by many prior works. The contributino tri-plane convolution also lacks complete analysis, such as discussion and comparison with similar modules that could fuse three axis information.

**Questions:**

1. What would happens if multiple objects instead of single object are encoded and fed into diffusion model in training? Would it not be able to generate realistic samples?
2. What does retargeting mean in this context? I could only get the meaning of outpaining and local editing.
3. How do the input noise and target tri-plane feature match each other when training the diffusion model on only one input shape? One input shape could only correspond to one tr-plane feature maps, while different noise is sampled in each iteration.

---

> ### Author Response · Authors · 2023-11-19
> **Response from authors**
>
> __Q: The application seems a bit narrow and limited.__
> __A:__ We stress that synthesizing new samples that resemble the input example has numerous downstream applications in the content production industry, and research on this topic dates back to the 1990s in vision and graphics [1][2]. The 2D counterpart of our method is the texture synthesis, which has been studied for a couple of decades (see discussion of related works in our revised paper) and many products are built on this idea. The 3D asset synthesis is highly desired, but remains challenging, in the gaming industry, film making, advertisement, to name a few. It is these high demands that motivate our work and justify the effort.
>
> [1] Efros, Alexei A., and Thomas K. Leung. "Texture synthesis by non-parametric sampling." ICCV, 1999.
> [2] Merrell, Paul. "Example-based model synthesis." I3D, 2007.
>
> __Q: Technical novelty. Discussion on tri-plane convolution.__
> __A:__ Though the network components (triplane autoencoder and latent diffusion model) are established techniques, our paper’s novelty lies in the adaptation for the single 3D shape generation task with limited receptive field and triplane-aware convolution to achieve experimentally verified strong results.
> In addition, we have an ablation study on tri-plane convolution to show its effectiveness (see Table 2 and Figure 12). If the reviewer has in mind a similar module that can fuse three axis information, we'd be happy to test it out.
>
> __Q: What would happen if training the model on multiple objects?__
> __A:__ We tried this in our early experiments. With multiple objects, the model is in effect trained to learn the patch distribution over all the input objects. As a result, the generated sample contains different patches from multiple inputs. It often looks unnatural if the multiple input objects have different global structures. However, this problem is out of the scope of this paper.
>
> __Q: What does retargeting mean in this context?__
> __A:__ It means generating new shapes whose sizes and aspect ratios are different from the input example.
>
> __Q: How do the input noise and target tri-plane feature match each other with only one input shape?__
> __A:__ It is correct that “one input shape could only correspond to one tr-plane feature map, while different noise is sampled in each iteration”. The training objective is essentially the typical diffusion model objective, with the expectation being solely over random noise (see Eqn. 6).

---

### Official Review · Reviewer_vrWR · 2023-11-01

**Soundness:** 3 good
**Presentation:** 3 good
**Contribution:** 2 fair
**Rating:** 6
**Confidence:** 4

**Summary:**

This paper introduces a diffusion model trained on a single 3D textured object. The goal here is to learn a generative model that can regenerate local patch wise patterns of the single object. The key idea is to train a path-based diffusion model on encoded latents of the single objects and use a triplane-base decoder to represent the 3D content. For the decoding part, the authors propose a novel triplane-aware convolutional architecture and 3D convolutional based encoder. The authors provide results on various 3D assets and provide an ablation study to support their claims.

**Strengths:**

The related work section covers relevant topics and helps to set the work in context.
The authors tackle a novel problem that I was not aware of before.
The results look plausible and claims are supported.

**Weaknesses:**

One point that remains unclear to me is how the triplane based representations can handle larger extensions of objects, e.g.  in the example of the building. How can the triplane represent those? Do you increase the size?

The authors claim that the model learns a distribution over patches. It is unclear to me how you choose the size of patches/ receptive field and how you identify that it is nor overfitting. I think there is more evaluation needed to show that it is not overfitting. Maybe an ablation on the receptive field size can lead to interesting insights.

**Questions:**

In my view, the diversity metrics measure the average distance between generated samples and it seems like the generated samples should be very different from each other to have an improved diversity. I’m unsure if this is a good metric. The goal you want to achieve is a good similarity of local details while having a diversity in the global structure. Maybe a patch-based distance vs. a global feature can help with that. Please comment on that.

---

> ### Author Response · Authors · 2023-11-19
> **Response from authors**
>
> __Q: How can the triplane represent larger extensions of objects, e.g. in the example of the building? Do you increase the size?__
> __A:__ We didn’t increase the triplane size for that example. With a size of 128, it is capable of representing all the objects that we tested.
>
>
> __Q: Patch size, receptive field and overfitting.__
> __A:__ We did conduct an ablation study on the receptive field size, shown in Figure 13 of the appendix. When the receptive field is too small, the generated shapes exhibit rich patch variations but fail to retain the global structure (low quality, high diversity). Conversely, when the receptive field is too large, the model overfits the input example and is unable to produce variations (high quality, low diversity). We choose the receptive field size to be 40% of the input size to strike a balance between quality and diversity.
>
>
> __Q: Diversity metrics. Maybe a patch-based distance vs. a global feature.__
> __A:__ It is correct that our goal is to “achieve a good similarity of local details while having a diversity in the global structure”. We measure this using both our quality and diversity metrics jointly. The geometry quality metric (G-Qual.) is based on distances between patch-level features, which quantifies the similarity of local details; the geometry diversity metric (G-Div.) is based on global intersection over union (IoU), which quantifies diversity of the global structure. The texture quality and diversity metrics are defined similarly.

---

### Official Review · Reviewer_Udpj · 2023-11-04

**Soundness:** 3 good
**Presentation:** 3 good
**Contribution:** 2 fair
**Rating:** 5
**Confidence:** 4

**Summary:**

This method introduces a way to generate new textured 3D shapes from a single example. To accomplish this, the input shape is represented using triplane feature maps, which encode signed distance as well as color. These feature maps are used to train an auto encoder, and a diffusion model is then trained on this auto encoder latent space. Then, by modifying the noise map, e.g., changing the resolution or masking regions, during the diffusion inference process, novel 3D shapes whose internal distribution resembles that of the original are generated.

**Strengths:**

The overall idea of using latent diffusion models to aid in learning the internal distribution of a 3D shape and the specific design choices in the method---e.g., the shape representation and the steps taken to prevent overfitting---are clever. The results shown also look nice and consistent with minimal artifacts.

**Weaknesses:**

It is mentioned several times that a unique benefit of the proposed method is that inputs and outputs meshes. This is not the case, as instead the 3D representation is triplanes encoding signed distance as well as texture color. While this can indeed be converted to a mesh using marching cubes, it's not fair to claim that the method outputs a mesh. This brings me to my main concern with the paper. While the authors cite and briefly discuss [Li et al. 2023], no qualitative or quantiative comparisons are provided, with the explanation that Li et al. do not produce parameterized meshes. While I agree that there are merits to the proposed representation, [Li et al. 2023] seems like the most natural point of comparison---many of the experiments are nearly identical to this in this work. For this reason, I cannot recommend acceptance as is.

Typos:

Abstract: "example as long" -> "example has long"
4: "The decoder first refine..." -> "The decoder first refines... gathers..."
Figure 3: replicating along which axis?

**Questions:**

How consistent/realistic are the normals generated in Figure 7? It would be interesting to show them used in relighting.

---

> ### Author Response · Authors · 2023-11-19
> **Response from authors**
>
> __Q: The claim that the method outputs a mesh.__
> __A__: In the revised version, we clarify that the proposed method does not directly output a mesh, but rather outputs a triplane representation that can be converted to a high quality textured mesh.
>
> __Q: Comparison to [Li et al 2023].__
> __A:__ We add the quantitative and qualitative comparison to Sin3DGen [Li et al 2023] in the revised paper. As reported in Table 1, our method achieves better scores in terms of geometry quality, texture quality and diversity. As shown in Figure 14 of the appendix, Sin3DGen [Li et al 2023], which is based on Plenoxels radiance fields, often produces broken surfaces and distorted textures. In contrast, our method is able to generate 3D shapes with high quality geometry and fine texture details.
>
> __Q: Typos.__
> A: Thanks for pointing these out. We have corrected them in the paper.

---

### Author Response · Authors · 2023-11-19
**General response**

We appreciate the reviewers finding our proposed method novel, presentation clear, and experimental results strong. We have addressed all concerns raised by the reviewers in our rebuttal and we hope the authors can engage in follow-up discussions if any other concerns still remain. We have submitted a revised version of the paper with revised texts are highlighted in blue color.  Specifically, in the revised version, the major changes include,
- we add quantitative and qualitative comparison to [Li et al 2023] (see Table 1 and Figure 14) (__reviewer Udpj__);
- we fixed a bug for computing texture quality score and updated Table 1 and Table 2 accordingly;
- we include more examples for retargeting and controlled generation Figure 15, and show failure cases in Figure 16 (__reviewer cGW2__);
- we clarify all the points that led to confusion (__reviewers Udpj, s6cL, cGW2__)

We address each reviewer's specific concerns in separate responses below.

---

### Meta-Review · Area_Chair_UGtD · 2023-12-06

**Metareview:**

This work presents a new approach for the challenging problem of synthesizing 3D textured structures based on observed exemplars.
In their approach they use a new architecture that can (validated experimentally) provide consistency with the input exemplar, while enabling spatial variations. The components of the proposed architecture has been introduced in prior work as pointed out by a reviewer. Therefore the novelty of the work is their specific employment in the proposed pipeline.

**Justification For Why Not Higher Score:**

The novelty of the work is the application of already-proposed-components in their model architecture.

**Justification For Why Not Lower Score:**

As reviewers point out , the proposed use of components is not a straight forward task and authors showed clear novelty and significant results for that contribution.

---

### Decision · Program_Chairs · 2024-01-16

Accept (poster)